The causal effects of circulating cytokines on sepsis: a Mendelian randomization study

Fang Weijun
Chai Chen
http://orcid.org/0000-0003-0026-6434 Lu Jiawei jwlu93@whu.edu.cn
Emergency Center, Zhongnan Hospital of Wuhan University, Wuhan, China , Wuhan , China
Uversky Vladimir
Electronic publication date: 2024 Feb 1
Publication date: 2024
Volume: 12
Electronic Location ID: e16860
Received 2023 Sep 21; Accepted 2024 Jan 9
Copyright: © 2024 Fang et al.
Copyright year: 2024
Copyright holder: Fang et al.
License: This is an open access article distributed under the terms of the Creative Commons Attribution License, which permits unrestricted use, distribution, reproduction and adaptation in any medium and for any purpose provided that it is properly attributed. For attribution, the original author(s), title, publication source (PeerJ) and either DOI or URL of the article must be cited.
License URL: https://creativecommons.org/licenses/by/4.0/

Keywords: Circulating cytokines, Sepsis, Mendelian randomization, Nerve growth actor, RANTES, Fibroblast grow factor

Funding: The authors received no funding for this work.

==============================
Background

In observational studies, sepsis and circulating levels of cytokines have been associated with unclear causality. This study used Mendelian randomization (MR) to identify the causal direction between circulating cytokines and sepsis in a two-sample study.

Methods

An MR analysis was performed to estimate the causal effect of 41 cytokines on sepsis risk. The inverse-variance weighted random-effects method, the weighted median-based method, and MR-Egger were used to analyze the data. Heterogeneity and pleiotropy were assessed using MR-Egger regression and Cochran’s Q statistic.

Results

Genetically predicted beta-nerve growth factor (OR = 1.12, 95% CI [1.037–1.211], P = 0.004) increased the risk of sepsis, while RANTES (OR = 0.92, 95% CI [0.849–0.997], P = 0.041) and fibroblast growth factor (OR = 0.869, 95% CI [0.766–0.986], P = 0.029) reduced the risk of sepsis. These findings were robust in extensive sensitivity analyses. There was no clear association between the other cytokines and sepsis risk.

Conclusion

The findings of this study demonstrate that beta-nerve growth factor, RANTES, and fibroblast growth factor contribute to sepsis risk. Investigations into potential mechanisms are warranted.

Introduction

Sepsis is a life-threatening organ dysfunction induced by the dysregulated host response to infection (Singer et al., 2016). Recent studies show that approximately 489 million people worldwide are diagnosed with sepsis each year, and approximately 11 million people lose their lives to sepsis (Rudd et al., 2020). Risk factors for sepsis include age (especially older adults and young children) and impaired immune function such as tumors, diabetes, severe trauma, and burns (Fathi, Markazi-Moghaddam & Ramezankhani, 2019). It is commonly believed that the development of sepsis involves both pathogenic microorganisms and the host’s immune status. When the host’s immune system detects a pathogenic invasion, immune cells are rapidly activated to produce and secrete a variety of cytokines to help the body clear the infection (Angus & van der Poll, 2013; Chousterman, Swirski & Weber, 2017; Fajgenbaum & June, 2020).

Cytokines are important immunomodulatory factors secreted during the immune response of the organism against infection (Chousterman, Swirski & Weber, 2017). Cytokines can be divided into two categories: pro-inflammatory positive cytokines stimulate the production of the systemic validation response, while anti-inflammatory cytokines inhibit the inflammatory response (Chaudhry et al., 2013). However, it is unclear whether cytokines promote or suppress the inflammatory response in sepsis. A large number of observational studies have also shown that the levels of both pro-inflammatory and anti-inflammatory cytokines in the blood are involved in the complex pathophysiological process of sepsis (Gogos et al., 2000; Tallon et al., 2020). Although satisfactory results have been obtained in animal models for blocking cytokine activity such as TNF-α (Dinarello, 1991), IL-17 (Wynn et al., 2016), and IL10 (Kalechman et al., 2002), the results in clinical trials on patients with sepsis are not clear (Lv et al., 2014; Rizvi & Gallo De Moraes, 2021). Therefore, the role of cytokines in sepsis needs to be further clarified.

Mendelian randomization (MR) analysis is a way to study the causal relationship of a specific exposure on the outcome using genetic variants associated with possible exposures as instrumental variables (Emdin, Khera & Kathiresan, 2017). In contrast to observational studies, the use of MR analysis helps avoid confounding factors and allows the discovery of causal effects because genetic variants are randomly inherited and alleles are not influenced by disease (Emdin, Khera & Kathiresan, 2017). To date, the Genome-Wide Association Study (GWAS) has identified tens of thousands of genetic variants associated with complex traits that can be used as instrumental variables for clarifying the causative factors of diseases. To further clarify the causal role of circulating cytokines in the development of sepsis, two-sample MR was used to analyze the causal relationship between circulating cytokines and sepsis risk.

Materials and Methods

Study design

This study was based on three major assumptions underlying MR studies. First, instrumental variables (IVs) must be strongly correlated with circulating cytokines. Second, instrumental variables are independent of confounding factors. Third, instrumental variables act on sepsis through circulating cytokines (VanderWeele et al., 2014). The flowchart of this study is shown in Fig. 1.

Figure 1 Overview of Mendelian randomization.

Genome-wide association summary data

Genome-wide association summary-level data was acquired for 41 circulating cytokines (CTACK, beta-nerve growth factor, vascular endothelial growth factor, macrophage migration inhibitory factor, TRAIL, tumor necrosis factor beta, tumor necrosis factor alpha, stromal-cell-derived factor 1 alpha, stem cell growth factor beta, stem cell factor, interleukin-16, ANTES, platelet-derived growth factor BB, macrophage inflammatory protein 1b, macrophage inflammatory protein 1a, monoline induced by gamma interferon, macrophage colony stimulating factor, monocyte chemoattractant protein-3, monocyte chemoattractant protein-1, interleukin-12p70, interferon gamma-induced protein 10, interleukin-18, interleukin-17, interleukin-13, interleukin-10, interleukin-8, interleukin-6, interleukin-1-receptor antagonist, interleukin-1-beta, hepatocyte growth factor, interleukin-9, interleukin-7, interleukin-5, interleukin-4, interleukin-2 receptor antagonist, interleukin-2, interferon gamma, growth-regulated protein alpha, granulocyte-colony stimulating factor, fibroblast growth factor basic, and eotaxin) from the Cardiovascular Risk in Young Finns study (FINRISK 1997 and 2002) (Ahola-Olli et al., 2017). Bio-Rad’s premixed 27-plex and 21-plex Human Cytokine Assay, plus a Bio-Plex 200 reader with Bio-Plex 6.0 software, were used to measure cytokine levels. A 1,000 Genomes Phase 1 reference haplotype was used for genotype imputation (Genomes Project et al., 2010). Single-variant associations were obtained by linear regression adjusting for age, sex, and body mass index between cytokine levels and single-nucleotide polymorphisms (SNPs) (Ahola-Olli et al., 2017).

The UK Biobank supplied summary-level GWAS data for sepsis, comprising 452,764 controls and 10,154 sepsis cases (Ponsford et al., 2020). The median age of all participants was 58 years, the median age of those suffering from sepsis was 60 years, and 54% of all participants were female (Ponsford et al., 2020). Global Burden of Disease (GBD) study codes were used to define sepsis based on International Classification of Disease (ICD)-9 and ICD-10 codes (Rudd et al., 2020).

The OpenGWAS database developed by the MRC Integrative Epidemiology Unit (IEU) (https://gwas.mrcieu.ac.uk/) provides a convenient way to acquire summary-level data. Details are available in Table S1 (Elsworth et al., 2020; Hemani et al., 2018b).

Selection of instrumental variables

All SNPs with a P < 5 × 10−6 were considered significant variants associated with phenotypes and were included in further sensitivity analyses to find potential causal effects (Long et al., 2023). Detailed information about sepsis-related SNPs are shown in Table S2. SNPs with R2 < 0.001 using linkage disequilibrium (LD) analysis were excluded. To avoid weak instrument bias, the F-statistic was calculated according to the formula F=R2(n−k−1)(1−R2)k, in which n is the sample size, k represents the number of SNPs, and R2 is the proportion of variance explained by the instrumental variants. An F-statistic value >10 was regarded as strong enough to avoid weak instrument bias. All IVs in this study had an F-statistic >10. IVs were selected based on 437 SNPs linked to 41 cytokines after screening. MR analyses were restricted to cytokines with IVs ≥ 3 in order to achieve a stable assessment (Burgess & Thompson, 2017).

Statistical analyses

The random-effects inverse variance weighting (IVW) method was used as the primary analysis to assess the association between circulating cytokines and sepsis risk (Pierce, Ahsan & Vanderweele, 2011), and several sensitivity analyzes were performed to assess the robustness of the preliminary analyses (Hemani, Bowden & Davey Smith, 2018a). First, the weighted median method was used to estimate the potential causal impact when IV violates standard assumptions. In addition, MR-Egger regression was performed to evaluate the presence of directional pleiotropy (Hemani, Bowden & Davey Smith, 2018a). Consistent causal effects were acquired across all three methods (IVW, weighted median, and MR-Egger) and P < 0.05 was used to define a significant causal effect. The heterogeneity of the instrument variable was determined using Cochran’s Q statistic (Hemani, Bowden & Davey Smith, 2018a). In addition, a sensitivity analysis was performed with the leave-one-out method. All MR analyses were performed using R (version 4.1.1, Vienna, Austria) with the “TwoSampleMR” package (version 0.5.6). P values < 0.05 were considered statistically significant.

Results

Of the 41 circulating cytokines, at least four available gene transcripts were identified with P < 5 × 10−6, LD (R2 < 0.1, distance > 500 kb). In total, 437 SNPs associated with the 41 cytokines were selected as IVs for this study, details of which are provided in Table S3.

The primary MR analysis by random effects IVW showed a significant causal effect of beta-nerve growth factor, RANTES, and fibroblast growth factor on the risk of sepsis (Figs. 2 and 3). The results showed that beta-nerve growth factor (OR = 1.12, 95% CI [1.037–1.211], P = 0.004) increased the risk of sepsis, while RANTES (OR = 0.92, 95% CI [0.849–0.997], P = 0.041) and fibroblast growth factor (OR = 0.869, 95% CI [0.766–0.986], P = 0.029) reduced the risk of sepsis.

Figure 2 Mendelian randomization estimates for the causal effect of circulating cytokines on sepsis risk.

An asterisk (*) denotes P < 0.05 (in bold). OR: odds ratio; CI: confidence intervals.

Figure 3 Primary results of MR analysis on sepsis.

MR analysis of cytokines on sepsis. The odds ratio (OR) was estimated using the random effect IVW method. An asterisk (*) denotes P < 0.05 (in bold). SE: standard error CI: confidence intervals.

The associations between 41 circulating cytokines and sepsis risk were analyzed using MR-Egger, weighted median, simple mode, and weighted mode (Table S2). The results when using the weighted median mode showed that beta-nerve growth factor (weighted median: OR = 1.112, 95% CI [1.005–1.230], P = 0.04.) increased the risk of sepsis, while RANTES (weighted median: OR = 0.894, 95% CI [0.802–0.997], P = 0.043) decreased the risk of sepsis. However, fibroblast growth factor (weighted median: OR = 0.854, 95% CI [0.721–1.011], P = 0.068) was not associated with the risk of sepsis with the weighted median method.

In the sensitivity analysis, there was no evidence for heterogeneity measured by Cochran’s Q-test (Qbeta-nerve growth factor = 4.776, P = 0.687; QRANTES = 4.965, P = 0.664; Qfibroblast growth factor = 4.850, P = 0.678). No pleiotropic effects were found by the MR Egger regression intercept (beta-nerve growth factor: intercept = −0.014, P = 0.681; RANTES: intercept = 0.012, P = 0.584; fibroblast growth factor: intercept = 0.010, P = 0.592). Details are shown in Table 1. The scatter plot of the effect estimates of IVs on beta-nerve growth factor, RANTES, and fibroblast growth factor and the risk of sepsis are presented in Figs. S1, S3 and S5, respectively.

Table 1 Heterogenity and pleiotropy analysis.

	Heterogenity	MR-Egger intercept	
Exposures	Q	Q_P value	Egger_intercept	SE	P value	
CTACK levels	10.741	0.465	0.007	0.014	0.638	
Beta-nerve growth factor levels	4.776	0.687	−0.014	0.032	0.681	
Vascular endothelial growth factor levels	7.061	0.631	−0.021	0.020	0.319	
Macrophage Migration Inhibitory Factor levels	4.582	0.801	0.017	0.016	0.342	
TRAIL levels	18.536	0.293	−0.004	0.009	0.691	
Tumor necrosis factor beta levels	4.239	0.237	0.018	0.014	0.322	
Tumor necrosis factor alpha levels	5.675	0.225	−0.027	0.015	0.178	
Stromal-cell-derived factor 1 alpha levels	9.083	0.335	0.002	0.012	0.851	
Stem cell growth factor beta levels	18.002	0.207	−0.003	0.011	0.766	
Stem cell factor levels	15.325	0.121	−0.013	0.016	0.432	
Interleukin-16 levels	6.444	0.598	0.013	0.015	0.410	
RANTES levels	4.965	0.664	0.012	0.021	0.584	
Platelet-derived growth factor BB levels	17.646	0.171	−0.013	0.011	0.268	
Macrophage inflammatory protein 1b levels	12.883	0.845	0.004	0.008	0.649	
Macrophage inflammatory protein 1a levels	2.804	0.833	0.020	0.021	0.391	
Monokine induced by gamma interferon levels	15.453	0.280	0.010	0.015	0.516	
Macrophage colony stimulating factor levels	4.539	0.806	−0.009	0.017	0.612	
Monocyte chemoattractant protein-3 levels	0.914	0.822	−0.003	0.029	0.932	
Monocyte chemoattractant protein-1 levels	11.304	0.662	−0.013	0.010	0.232	
Interleukin-12p70 levels	19.607	0.105	−0.008	0.010	0.401	
Interferon gamma-induced protein 10 levels	6.051	0.642	−0.012	0.013	0.394	
Interleukin-18 levels	11.363	0.878	0.002	0.009	0.860	
Interleukin-17 levels	18.474	0.071	0.030	0.018	0.125	
Interleukin-13 levels	8.215	0.314	−0.014	0.017	0.434	
Interleukin-10 levels	15.142	0.127	−0.014	0.012	0.252	
Interleukin-8 levels	1.225	0.747	0.001	0.014	0.960	
Interleukin-6 levels	6.024	0.537	−0.015	0.014	0.323	
Interleukin-1-receptor antagonist levels	6.355	0.608	0.017	0.017	0.338	
Interleukin-1-beta levels	1.192	0.879	−0.020	0.019	0.371	
Hepatocyte growth factor levels	16.328	0.022	−0.029	0.028	0.338	
Interleukin-9 levels	11.663	0.070	0.019	0.029	0.552	
Interleukin-7 levels	6.919	0.806	−0.010	0.016	0.538	
Interleukin-5 levels	4.865	0.301	−0.017	0.025	0.540	
Interleukin-4 levels	8.296	0.600	0.012	0.011	0.311	
Interleukin-2 receptor antagonist levels	1.875	0.966	−0.001	0.012	0.951	
Interleukin-2 levels	9.779	0.369	0.007	0.012	0.557	
Interferon gamma levels	13.637	0.325	0.014	0.012	0.275	
Growth-regulated protein alpha levels	13.213	0.105	0.010	0.022	0.672	
Granulocyte-colony stimulating factor levels	8.435	0.392	−0.017	0.011	0.162	
Fibroblast growth factor basic levels	4.850	0.678	0.010	0.017	0.592	
Eotaxin levels	10.516	0.724	0.005	0.012	0.661	
Note: Q-value: Cochran’s Q statistic, SE: standard error.

Plots of the leave-one-out analysis demonstrated the causal link between beta-nerve growth factor (shown in Fig. S2), RANTES (Fig. S4), and fibroblast growth factor (Fig. S6).

Discussion

Previous studies suggest a significant relationship between circulating cytokines and sepsis (Chaudhry et al., 2013; Chousterman, Swirski & Weber, 2017; Tallon et al., 2020), but the causal relationship remains unclear. In the present study, the causal relationship between 41 cytokines and sepsis was investigated using the two-sample MR method. The results showed that beta-nerve growth factor increased the risk of sepsis, while RANTES and fibroblast growth factor decreased the risk of sepsis.

Sepsis is a dysregulated immune response of the body to fight infection (Singer et al., 2016). Research into sepsis has found that during the initial stages, the immune system is suppressed. This may be due to cytokines produced by immune cells to eliminate the pathogen (Chaudhry et al., 2013). However, the sudden uncontrolled release of a large number of cytokines, known as cytokine storms, can lead to multiple organ failure and ultimately, death (Fajgenbaum & June, 2020). It is not currently clear which immune cells and cytokines may be involved in this pathological hyperinflammatory response in sepsis (Fajgenbaum & June, 2020). In observational studies, multiple cytokines have been found to be associated with the prognosis of sepsis, including both pro-inflammatory cytokines, such as TNF-α, IL-1, IL-6, IL-12, IFN-γ, and MIF, as well as cytokines that suppress inflammation, such as IL-10, TGF-β, and IL-4 (Chaudhry et al., 2013; Chousterman, Swirski & Weber, 2017; Wang, Zhao & Wang, 2018). Although in animal sepsis models, the use of specific neutralizing antibodies blocking the activity of TNF-α (Lv et al., 2014) and IL-10 (Kalechman et al., 2002) significantly improved the prognosis of sepsis, neutralizing antibodies to TNF-α in clinical trials did not yield satisfactory results (Lv et al., 2014).

The beta-nerve growth factor (NGF) is a substance that promotes the growth, development, differentiation, and maturation of central and peripheral neurons, maintains the normal function of the nervous system, and accelerates the repair of the nervous system after injury (Wiesmann & de Vos, 2001). NGF is widely distributed in all tissues and organs of the body, and studies have found that it is also able to influence the activity of immune cells and modulate the function of the immune system (Aloe et al., 2012; Lambiase et al., 2004). In animal experiments, NGF levels were significantly higher in septic rats compared to the control group 24 h after inducing sepsis with LPS (lipopolysaccharide). The same study also discovered an increase in apoptosis in the liver and lung tissues of the sepsis group, with a more pronounced effect on intestinal tissues (Bayar et al., 2010). Following anti-NGF treatment 24 h after sepsis induction, Bcl-2 expression increased, and Bax expression decreased in mouse tissue, resulting in a reduction of apoptosis. This suggests that NGF may promote the progression of sepsis by inducing apoptosis (Bayar et al., 2010). In vitro experiments demonstrate that NGF triggers apoptosis by activating stress-activated protein kinase/c-Jun amino terminal kinase within the NF-κB signaling pathway (Kuner & Hertel, 1998). Recent studies have found significantly higher concentrations of NGF in the blood of patients with sepsis than in the healthy population (Jekarl et al., 2019). Activation of the NGF signaling pathway in elderly patients with sepsis was associated with a poor prognosis in one clinical study (Vieira da Silva Pellegrina et al., 2015). These results suggest that NGF is involved in the immune inflammatory response of the body against sepsis, and the present study also confirms that NGF increases the risk of sepsis. Although there is clinical evidence associating NGF with sepsis, further research is required to clarify its precise mechanism.

RANTES, also known as CCL5, is a chemotactic cytokine, recruiting leukocytes to the site of inflammation (Appay & Rowland-Jones, 2001). The chemokine CCL5 is mainly expressed by T-cells and monocytes and is abundantly expressed by epithelial cells, fibroblasts, and thrombocytes (Hinrichs et al., 2022). One study found that low levels of RANTES are associated with mortality in children with cerebral malaria (John et al., 2006), suggesting that RANTES has a positive role in counteracting infection. However, sepsis animal studies found that administration of CCL5 increased sepsis-induced lethality in wild type mice, whereas neutralization of CCL5 improved survival (Ness et al., 2004), consistent with the results of the present study. One animal study showed that administering CCL5 to mice caused an increase in mortality in wild-type mice with sepsis, whereas neutralizing CCL5 resulted in improved survival. Further investigation revealed that CCR1/CCL5 receptor-ligand activated NF-κB interaction led to the production of damaging levels of IFN-gamma and MIP-2 (Ness et al., 2004). Another study found that sepsis activates Rac1 in platelets, resulting in the secretion of CCL5 by the platelets. CCL5 triggers alveolar macrophages to express CCR1 and CCR5 whilst secreting CXCL2. As a result, neutrophils gather within the lung tissue, leading to the formation of tissue pulmonary edema and injury (Hwaiz et al., 2015). Reduced CCL5 may thus assist in managing the progression of sepsis.

Fibroblast growth factors (FGF) are a family of cell signaling proteins produced by macrophages, involved in cell proliferation, cell migration, cell differentiation, and angiogenesis (Yun et al., 2010). A total of 23 FGF molecules have been identified so far (Ornitz & Itoh, 2001; Yun et al., 2010). A recent study found that FGF5 protects against myocardial injury caused by sepsis by inhibiting CaMKII/NF-κB signaling (Cui et al., 2022). FGF2 inhibits coagulation activity in septic mice by inhibiting the AKT/mTOR/S6K1 pathway, alleviating lung and liver damage and improving survival (Sun et al., 2023). FGF2 also inhibits LPS-induced endothelial cell injury and macrophage inflammation through the AKT/P38/NF-κB signaling pathway, alleviating pulmonary capillary leakage and lung injury and increasing survival rates in mice (Pan et al., 2020). In a study of sepsis patients, those with low concentrations of FGF21 in the blood had a better prognosis compared those with high concentrations (Li et al., 2018). However, in community-dwelling adults, higher FGF23 concentrations were not independently associated with a higher risk of sepsis (Gariani et al., 2013). Thus, the function of FGF in sepsis remains unclear and needs further study.

Compared to traditional observational studies, this MR study was free from the effects of confounding factors and reverse causality, allowing a credible assessment of the causal relationship between circulating cytokines and sepsis risk. However, this study has some non-negligible limitations. First, the data was sourced mainly from the European population, so the findings are not directly applicable to other populations. Second, 5 × 10−6 instead of 5 × 10−8 was used as the significance P-value in this study. Although this linear P-value threshold increases the number of available IVs for MR studies, it may also produce some bias.

Conclusions

This study found that beta-nerve growth factor, RANTES, and fibroblast growth factor are associated with the risk of sepsis, with beta-nerve growth factor increasing sepsis risk and RANTES and fibroblast growth factor decreasing sepsis risk. This finding may help explain the pathophysiological mechanisms of sepsis occurrence, but more in-depth studies are needed to further clarify this finding and to discover the underlying biological mechanisms for sepsis that may provide new directions and ideas for sepsis treatment.

Supplemental Information

Supplemental Information 1 Scatter plot of beta-nerve growth factor levels.

Click here for additional data file.

Supplemental Information 2 Leave-one-out plot of beta-nerve growth factor levels.

Click here for additional data file.

Supplemental Information 3 Scatter plot of RANTES levels.

Click here for additional data file.

Supplemental Information 4 Leave-one-out plot of RANTES levels.

Click here for additional data file.

Supplemental Information 5 Scatter plot of Fibroblast growth factor basic levels.

Click here for additional data file.

Supplemental Information 6 Leave-one-out plot of Fibroblast growth factor basic levels.

Click here for additional data file.

Supplemental Information 7 Data source and description.

Click here for additional data file.

Supplemental Information 8 Characteristics of the genetic instrument variables for the cytokines at the genome-wide significance level.

Click here for additional data file.

Supplemental Information 9 MR analysis of risk of sepsis.

Click here for additional data file.

Supplemental Information 10 STROBE-MR Checklist.

Click here for additional data file.

Supplemental Information 11 Codes.

Click here for additional data file.

Additional Information and Declarations

Competing Interests

Author Contributions

Data Availability

The authors declare that they have no competing interests.

Weijun Fang performed the experiments, analyzed the data, prepared figures and/or tables, and approved the final draft.

Chen Chai analyzed the data, prepared figures and/or tables, and approved the final draft.

Jiawei Lu conceived and designed the experiments, performed the experiments, analyzed the data, prepared figures and/or tables, authored or reviewed drafts of the article, and approved the final draft.

The following information was supplied regarding data availability:

The OpenGWAS database developed by the MRC Integrative Epidemiology Unit(IEU) (https://gwas.mrcieu.ac.uk/) provides a convenient way to acquire the summary-level data. Details are available in Table S1. The codes are available in the Supplemental File.

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
