# Peer review of "The causal effects of circulating cytokines on sepsis: a Mendelian randomization study"

_PeerJ, doi:10.7717/peerj.16860_

## Round 0.1 · original submission · Major Revisions

Please address the concerns of both reviewers and amend the manuscript accordingly.

**Language Note:** The review process has identified that the English language must be improved. PeerJ can provide language editing services - please contact us at copyediting@peerj.com for pricing (be sure to provide your manuscript number and title). Alternatively, you should make your own arrangements to improve the language quality and provide details in your response letter. – PeerJ Staff

·

Basic reporting

Authors have done a commendable research work in attempting to understand the relationship between cytokines and sepsis. The research manuscript is seemingly comprehensive, however, some major revisions are advised.

1. The introduction was well constructed but has not pointedly mentioned the research question across its structure. The obvious question is: Are cytokines good or bad when it comes to sepsis? This question should be described in the second paragraph somewhere between lines 45 and 46.

2. The use of possessive pronoun "we" across the manuscript should be removed. For example in Line 73 "We acquired genome-wide..." can be replaced with "A genome-wide association was acquired..."

Experimental design

The Materials and methods section is the suitable region to critically justify the selection of the MR methodology, especially its advantages over others. The introduction should briefly review other techniques that are commonly used.

Validity of the findings

A deeper discussion into the biochemistry of the results should be made available. For example, Lines 147-148 states "MR Egger and Cochran's Q test, and we did not find the presence of heterogeneity and polymorphism..."

The question is WHY? WHY was the heterogeneity not found? The potential biochemistry behind every result should be critically reviewed in the discussion section with reference to already published studies.

Reviewer 2 ·

Basic reporting

The English language is the most important issue for this manuscript and should be improved to ensure that an international audience can clearly understand the authors' text and accurately report their results. Some examples where the language could be improved include:
1. line 86, 116, 118, and 120, 121, 131 etc, wrong period (before the reference) or comma or blank was used
2. sentences in line 92-93, 123-125, 147 - 149, 188-189 are not clear to understand, and sentence in line 163 - 168 is too long and hard to follow
3. line 118 and line 120: different tenses were used. Please unify the tense used throughout the manuscript
I suggest the authors have a colleague who is proficient in English and familiar with the subject
matter review this manuscript, or contact a professional editing service.

Other issues are about the references and figures.
The reference in line 34 mentioned "In 2017, an estimated 48·9 million incident cases of sepsis were recorded worldwide and 11·0 million sepsis-related deaths were reported.", while the authors wrote "...49 million people worldwide are diagnosed with sepsis each year," Please do citation accurately.
Figure 2: please provide caption for "OR" and "CI" with this figure
Figure 3: please provide caption for "SE" etc with this figure;

Experimental design

I thank you for providing the raw data and detailed method, however the "Materials & Method" part needs to be improved in the following ways:
1. please revise any language issue as suggested above, for example in line 116 please don't put a period before the reference
2. please correct any wrong expressions: for example line 102: "P<5*10-6" and line "R2" (there is no "R2" in the formula in line 107)
3. a minor comment: in line 91, SNP first comes out but no full name is given, but later in line 105 full name is given to SNP, please correct

Validity of the findings

no comment

---

## Round 0.2 · Minor Revisions

Please address the remaining concerns of the reviewers and amend the manuscript accordingly.

·

Basic reporting

Authors have made considerable changes to the manuscript that has improved its technicality. In addition, there is now a pointed declaration on the study objective and the results obtained: no clear correlation between cytokines and sepsis.

However, some minor revisions are suggested.

Experimental design

Figure 1 describes Mendelian Randomization structure. A graphical abstract that shows in conciseness, the study methodology will be helpful.

Validity of the findings

Results are all in very large tables with high number of rows. It is suggested that some of these tables can be converted to graphical figures for easier comprehension of the study results.

Reviewer 2 ·

Basic reporting

The authors already modified or revised the old manuscript largely as I have suggested in my first review. The English language is good for publication and the method part provides enough details. I suggest the authors to proof-read the manuscript before the publication.
Minor comment:
please explain what is the meaning of "LPS" in lines 181 and 218.

Experimental design

no comment

Validity of the findings

no comment

Additional comments

no comment

---

## Round 0.3 · accepted · Accept

All concerns of the reviewers were adequately addressed, and the revised manuscript is acceptable now.